

# Observation of an optical anisotropy in the deep glacial ice at the geographic South Pole using a laser dust logger

Martin Rongen[1], Ryan Carlton Bay[2], and Summer Blot[3]

[1]RWTH Aachen University, Institute for Particle Physics III B, 52074 Aachen, Germany
[2]Department of Physics, University of California, Berkeley, CA, USA
[3]DESY, D-15738 Zeuthen, Germany

**Correspondence:** Martin Rongen (rongen@physik.rwth-aachen.de)

**Abstract.** We report on a depth-dependent observation of a directional anisotropy in the recorded intensity of back-scattered light as measured by an oriented laser dust logger. The measurement was performed in a drill hole at the geographic South Pole, about a kilometer away from the IceCube Neutrino Observatory. The drill hole remains open for access, after the SPICEcore collaboration had retrieved a 1751 m ice core. We find the measured anisotropy axis of $126 \pm 3°$ to be compatible with the local flow direction. The observation is discussed in comparison to a similar anisotropy observed in data from the IceCube Neutrino Observatory and is able to dismiss Mie scattering based explanations in favor of a birefringence based scenario. In the future, the measurement principle, when combined with a full-chain simulation, may be used to provide a continuous record of fabric properties along the entire depth of a drill hole.

## 1 Introduction

The viscosity of an individual ice crystal strongly depends on the direction of the applied strain. As a hexagonal crystal, ice will most readily deform as shear is applied orthogonal to the c-axis, which leads to slip of the basal planes (Petrenko and Whitworth (2002)). Thus, individual grains elongate, with the major axis being aligned perpendicular to the c-axis.

In large-scale systems, such as glaciers or ice sheets, ice is compressed under its own weight and as a result flows away from the accumulation region. This leads to preferential c-axis orientations (see Alley, 1988), most commonly girdle fabrics, where c-axis are predominantly found on a plane, with the plane's normal vector being aligned with the flow direction. Ice fabric can not only be observed through macroscopic imaging of ice cores (Weikusat et al., 2016), but also leads to a directionality in the propagation of mechanical and electromagnetic radiation, in principle allowing for remote-sensing of the ice fabric.

The mechanical anisotropy of ice means that the speed of sound depends on the fabric realization. This has for example been derived and measured by Kluskiewicz et al. (2017). Ice crystals are also a birefringent material, with any incoming electromagnetic radiation being separated into an ordinary and extra-ordinary ray of opposite polarization with respect to the c-axis, and which propagate with different refractive indices. This is classically observed as a direction-dependent delay in the propagation





of radio waves, as for example described by Fujita et al. (2006).

Recently, as part of ice calibration measurements for the IceCube Neutrino Observatory (Aartsen et al., 2017), Chirkin (2013) described the observation of an optical anisotropy, where about twice as much light is observed along the glacial flow
5    axis versus orthogonal to the flow axis, at a receiver 125 m away from an isotropic emitter. The effect was originally modelled as a direction dependent modification to Mie scattering quantities, either through a modification of the scattering function as proposed by Chirkin (2013) or through the introduction of a direction dependent absorption as introduced by Rongen (2019). As also shown by Rongen (2019), both parameterizations lack a thorough theoretical justification and resulted in an incomplete description of the IceCube data.

As the wavelength of ∼400 nm employed in the IceCube studies is significantly smaller then the average grain size, the effect is challenging to derive from first principles. First attempts have been made by Chirkin and Rongen (2019) by attributing the effect to the cumulative diffusion that a light beam experiences as it is refracted or reflected on many grain boundary crossings in a birefringent polycrystal with a preferential c-axis distribution. In this scenario the diffusion is found to be strongest when
photons initially propagate along the flow and smallest when initially propagating orthogonal to the flow. In addition photons are, on average, deflected towards the flow axis. For crystal realizations where the deflection outweighs the additional diffusion, the photon flux along the flow axis will continuously increase with distance.

We add to the body of anisotropy observations by providing the first direction-dependent measurement of the intensity of
back-scattered, optical light returning to the oriented dust logger deployed down a glacial bore hole. If the anisotropy is caused by Mie scattering a reduced return signal is expected when the light source points along the flow, while more light is expected to return in case of the birefringence and absorption explanations.

## 2    The oriented dust logger

The dust logger, as sketched in Figure 1 and introduced by Bramall et al. (2005), consists of a 404 nm laser line source, emitting
a 2 mm thin, horizontal fan of light about 60° across. A small fraction ($10^{-10}$ to $10^{-6}$) of all emitted photons is back-scattered or reflected and returns to the bottom section of the dust logger where a 1" Hamamatsu photon-counter module is located. Scattering and absorption on air bubbles, soot and other impurities as described by Mie scattering theory is traditionally thought to be the dominant contribution to the return signal. However, taking into consideration the findings of Chirkin and Rongen (2019), diffusion on grain boundaries may also contribute non-negligibly to the signal.

The intensity of the light source can be adjusted throughout the logging process. To avoid stray light contamination from reflections on the hole-ice interface, multiple sets of black nylon baffles are attached to the side of the pressure housing. These



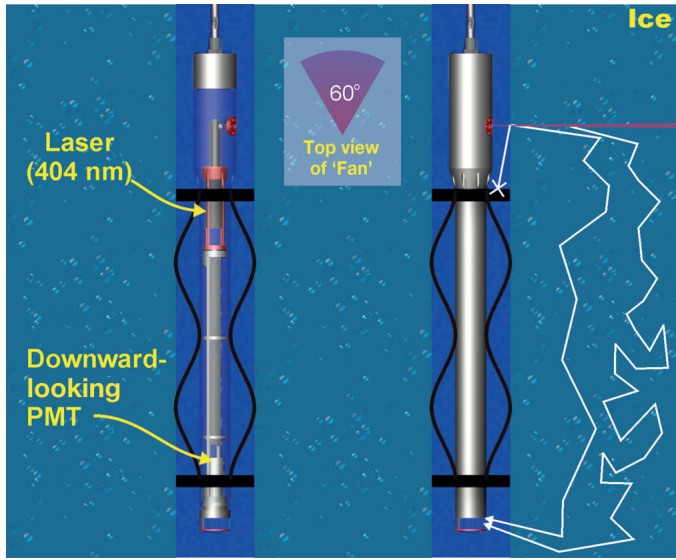

**Figure 1.** Sketch of the laser dust logger. Light emitted by a 404 nm diode laser in a 60° horizontal fan can only reach the photon counter below through scattering on nearby impurities.

also sweep ice crystals and debris out of the source beam. Spring-loaded calipers keep the logger centered in the hole.

The depth of the logger is monitored through the cable payout and on-board pressure sensors. During offline analysis multiple logs from the same site are further aligned to achieve centimeter depth precision, using characteristic features such as
5    volcanic lines, as described by Aartsen et al. (2013).

This device has previously been deployed in West Antarctica, East Antarctica and Greenland. Due to excellent imaging properties, deployments down the water-filled drill holes of the IceCube Neutrino Observatory resulted in one of the highest resolution particulate stratigraphies of any glacier available to-date, as described by Aartsen et al. (2013).

To measure a potential directionality of the return signal relative to the direction of the emitted light fan, an optional extension consisting of an Applied Physics Systems Model 547 Directional Sensor[1] has been fitted to the top of the logger. By measuring the local magnetic field it deduces the absolute orientation with an azimuthal accuracy of $\pm 1.2°$ for latitudes $< \pm 40°$. For our application at the geographic South Pole we estimate the azimuthal accuracy to be $\pm 3°$.

---

[1]https://www.appliedphysics.com/_main_site/wp-content/uploads/model-547-micro-orientation-sensor.pdf



## 3 SPICEcore deployments

The South Pole Ice Core, SPC14 (see Casey et al., 2014), was drilled by the SPICEcore project in 2014–2016 at a location 2.7 km from the Amundsen–Scott station, using the Intermediate Depth Drill designed and deployed by the U.S. Ice Drilling Program (IDP) (Johnson et al., 2014). It reached a final depth of 1751 m (Winski et al., 2019), surpassing the original 1500 m

goal.

The core has been retrieved in 2 m segments with a diameter of 98 mm. The resulting 126 mm diameter drill hole was filled with Estisol-140 anti-freeze liquid and has been preserved for future logging access.

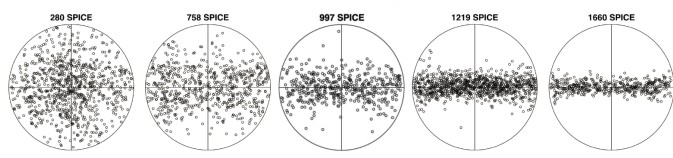

**Figure 2.** Depth development of c-axis distributions (Lambert azimuthal equal-area projections) measured in SPICEcore. Depths (in meters) are indicated atop each diagram. They show a clean girdle fabric below ∼1000 m. Adapted from: Donald E. Voigt (2017)

Unlike most ice coring sites, the Geographic South Pole is not near an ice divide, but rather on a flank site with a local flow velocity of 10 m per year. The associated accumulation site is believed by Lilien et al. (2018) to be Titan Dome, meaning that the ice has been transported over roughly 200 km. The experienced strain below ∼ 800 m has resulted in a very prominent and continuously strengthening girdle fabric as measured by Donald E. Voigt (2017). Figure 2 shows example c-axis distributions from the SPC14 ice core at various depths.

The oriented dust logger has been deployed down the SPICEcore hole twice during the 16/17 season, both times using the Intermediate Depth Logging Winch provided by the IDP. Due to a limited available cable length, it was only able to reach a depth of 1577 m of the 1751 m cored. During the first log the laser intensity was not yet optimized, leading to saturated and thus unusable data above 1000 m.

Two further deployments down to ∼1700 m were performed during the 18/19 season. Due to mechanical problems with the winch cable payout, depth readings from the winch itself are inaccurate. Only one deployment could be depth aligned to the required precision using characteristic features as previously discussed. This deployment includes an additional round-trip between 1354-1703 m. Table 1 summarizes the properties of the logs used for the measurements presented here.

As shown in Fig.3, the logger rotates as it descends and ascends the hole, mainly due to the residual twist in the logging cable. On ascent, as the cable is pulling the tool up, it undergoes a smooth rotation of slightly varying angular velocity. On




| Log | Depth [m] | Note |
|---|---|---|
| Down1 | 130 - 1577 | Saturated above 1000 m |
| Up1 | 1577 - 130 | Saturated above 1000 m |
| Down2 | 130 - 1580 | - |
| Up2 | 1580 - 130 | - |
| Down3-Leg1 | 130 - 1703 | - |
| Up3-Leg1 | 1703 - 1354 | - |
| Down3-Leg2 | 1354 - 1704 | Near identical orientations to Down3-Leg1 |
| Up3-Leg2 | 1704 - 130 | - |

**Table 1.** Summary of usable logs obtained within the 16/17 and 18/19 logging seasons.

descent the logger sinks under its own weight and the rotation is not continuous. The most likely explanation is that the logger is repeatedly stuck on the wall before slipping.

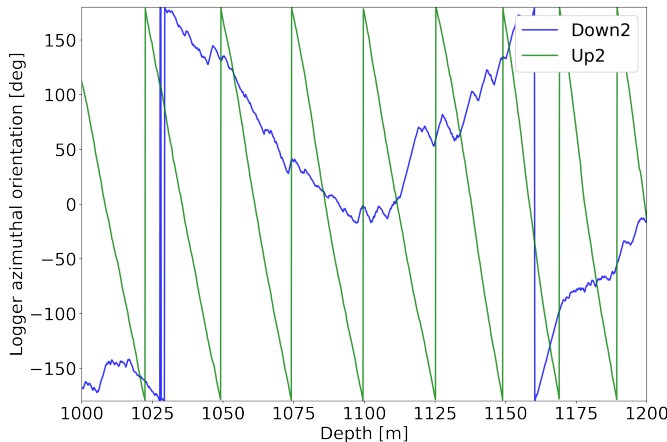

**Figure 3.** Rotation of the tool descending and ascending the bore hole. A smooth and continuous rotation of slightly varying angular velocity is observed when ascending. During descent the rotational movement is erratic.





## 4 Anisotropy signature

The data obtained from the oriented dust logger consists of orientation, depth and optical return signal measurements at 10 ms intervals. At the usual deployment speed this is equivalent to a sampling distance of ∼2.5 mm.

While the photomultiplier is located ∼850 mm below the laser light source, the depth resolution, as measured by Bramall et al. (2005) as the smearing of an ash layer, is dominated by the vertical extent of the laser beam and is no worse than a few mm. This allows for a continuous record of optical properties down the entire depth of a drill hole at the vertical resolution of less than a year of deposition, assuming an annual layer thickness of 1-2 cm in the deep ice as reported by Aartsen et al. (2013).

Above the transition region of air bubbles to craigite (Miller, 1969) at 700-1300 m the return signal is dominated by scattering on air bubbles. Below, the return signal is primarily proportional to the concentration of impurities contributing to scattering. The resulting high resolution stratigraphy is exemplified in Fig. 4.

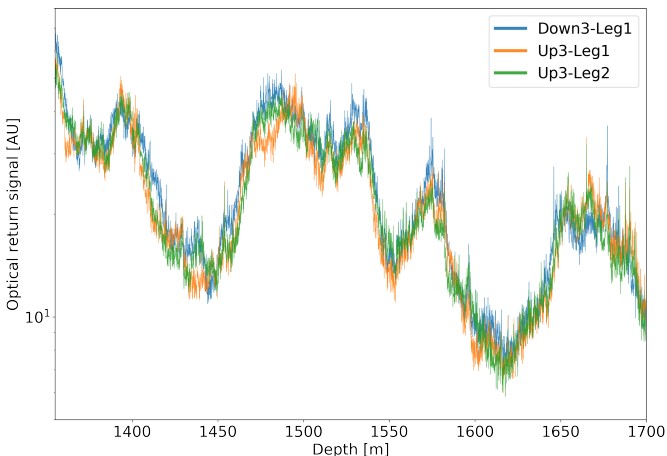

**Figure 4.** Example detail of optical stratigraphies obtained by logging the same depth range three times.

This figure also shows that the optical return signals at the same depths are not consistent between logs. Instead the signal depends on the absolute orientation of the logger. We extract this anisotropy signature by taking the ratio of two logs. Usually,
the ratios of raw data are on average non-unity and show slow, continuous variations. These global offsets are corrected using a second-degree polynomial fitted to the ratio.

Example ratios for ∼100 m depth slices and after fitting and correcting the offset for each depth slice are given in Fig. 5. When the device's orientation between logs is out of phase, pointing in different directions at the same depth, the ratio in
these examples becomes as large as 1.5. When the logs are in phase, the observed intensities are equal and thus the ratio is unity.



Analysis of the 18/19 logging season reveals that two logs, *Down3-Leg1* and *Down3-Leg2*, exhibit strongly correlated orientations. As exemplified in Fig. 6, the orientations of the two descending segments in the depth range between 1354 m and 1703 m show near identical orientations, suggesting that the rotation of the tool may have been governed by the hole geometry itself. As a result, the ratio of the two logs is consistent with unity where the logs align and the spread of 2% standard deviation indicates the typical short term intensity fluctuations seen in the measurement.

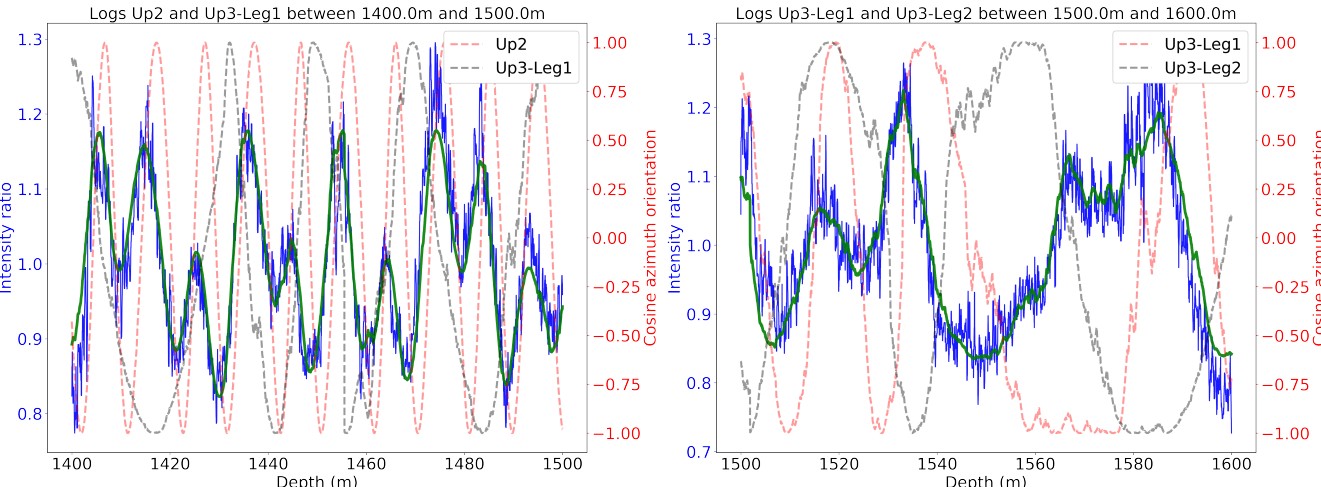

**Figure 5.** Example intensity ratio fits in ∼100m depth slices. The red and black dotted lines denote the orientations of the two used logs. (The orientation is defined as the cosine of the azimuth angle.) Blue is the intensity ratio. Green is the fitted intensity ratio using eq. 1.

In the following analysis, log Down3-Leg2 is excluded from the analysis as it is fully correlated with Down3-Leg1. The number of remaining usable ratios at each depth range available from $N$ logs is given by the binomial coefficient $\binom{N}{2}$ and varies between 3 and 21.

Lacking a mature first-principle explanation and simulation of the anisotropy effect and experimental setup, the ratios have instead been found to be well-described by the following empirical relationship:

$$\text{intensity ratio} = \frac{1 + a \cdot \cos\left(2 \cdot (\alpha_1 - \phi)\right)}{1 + a \cdot \cos\left(2 \cdot (\alpha_2 - \phi)\right)}. \tag{1}$$

Here $\alpha_1$ and $\alpha_2$ denote the azimuthal orientations of the logger during the two logs. $\phi$ denotes the azimuthal phase angle of the anisotropy effect, also called anisotropy axis and is limited to $0° - 180°$ . $a$ is a measure of the strength of the observed effect.

The orientations and the intensity ratio are given by the dust logger data. The free parameters $a$ and $\phi$ can be determined by fitting eq. 1 to the data from a given depth range.





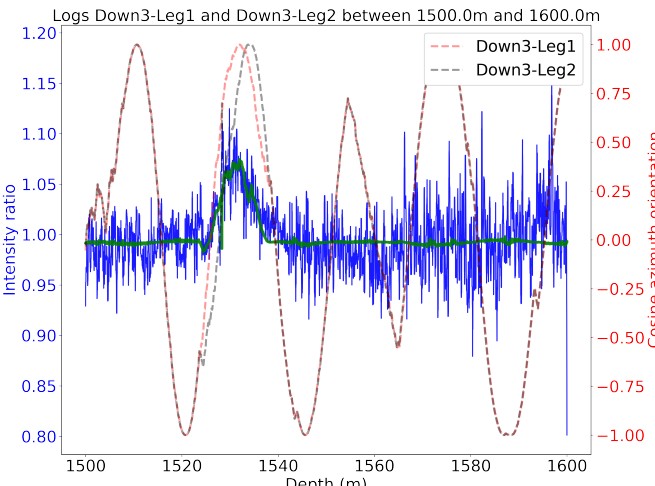

**Figure 6.** The intensity ratio of two log segments with identical orientations is unity and shows the typical spread of the data.

Note that the chosen relationship, while being very robust and easy to extract, implicitly assumes that the anisotropy causes a relative modulation to the total signal. In case the signal is additive on top of a contribution caused by Mie scattering on impurities, as would be expected from a fabric driven birefringence scenario, the derived strength parameter $a$ can not directly be interpreted as the strength of the underlying effect. For example, assuming an overall constant return signal from the anisotropy, the strength parameter $a$ would be seen to increase as the overall return signal decreases as a function of depth.

## 5 Depth evolution

To study the depth evolution of the anisotropy signature the data is binned into $100\,\mathrm{m}$ slices. While this only allows for a rather coarse depth resolution, it ensures that at least a few rotations are seen in each ratio. Otherwise the correction polynomial could bias the signal introduced by the anisotropy and the fit would not be able to reliably determine the phase and amplitude of each log. The systematic shift introduced by the correction polynomial was further accessed by varying its degree between 1 and 3 and was found to be below the error on the mean, which is introduced below, for all depth-slices.

In the future, a finer depth resolution may experimentally be achieved by not relying on the natural rotation of the logger as it is deployed but by artificially inducing a fast rotation.

The depth evolution of the fitted anisotropy axis $\phi$ and strength $a$ as a function of depth are seen in Fig. 7. The spread of these fitted quantities between different ratios far outweighs the statistical error of the fit of each ratio. The errors on the means of each depth bin are thus constructed from the standard deviation of all ratios. While $\binom{N}{2}$ ratios can be constructed, only $N-1$



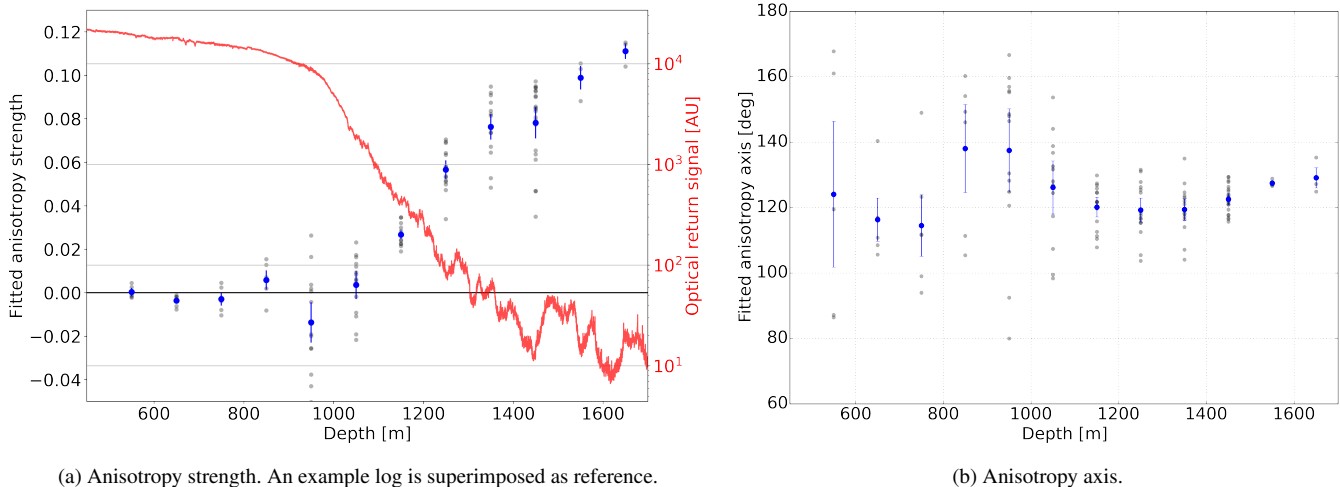

(a) Anisotropy strength. An example log is superimposed as reference.

(b) Anisotropy axis.

**Figure 7.** Depth-dependence of fitted quantities. Gray dots denote values obtained from individual ratios. The blue markers indicate the average at each depth.

are statistically independent. The error on each mean is thus given as $\sigma_{\mathrm{mean}} = \sigma/\sqrt{N-1}$.

Individual ratios are seen to yield consistent anisotropy axes in the deep ice. In the shallow ice above ∼1100 m, where the mean strength of the observed anisotropy signal vanishes, the phase angle is badly constrained and varies widely between individual ratios. The per-depth average axes are compatible with the global average of $126\pm1$ (stat) °. Considering the 3° systematic uncertainty of the orientation sensor, this direction is in good agreement with the local ice flow direction as measured by Lilien et al. (2018) as well as the optical axis of 126° or 130° as fitted by Chirkin (2013) and Rongen (2019) respectively, both using IceCube data.

No anisotropy signatures are seen above 1100 m, with the observed strength parameter continuously increasing in the deeper ice. It is currently unclear which fraction of the increase in observed anisotropy strength versus depth is caused by the continuously stronger girdle fabric or by the decrease in overall scattering. However, no anisotropy signal is observed at 1000 m where the girdle fabric is already clearly developed (see Fig. 2) but bubbles still dominate the scattering at this depth. Therefore, we suspect that the anisotropy signature is smeared out to some extent due strong local diffusion.

## 6 Conclusions

We have presented the first direction-dependent measurement of the intensity of back-scattered, optical light in deep glacial ice. The measurement has been performed using an oriented dust logger deployed down the SPICEcore drill hole.

Below ∼1100 m a consistent increase in received intensity is observed when the laser is aligned with the local flow axis. This is consistent with the birefringence explanation offered by Chirkin and Rongen (2019) where more diffusion is observed along the flow, thus leading to a higher return intensity] and inconsistent with the previous explanation given by Chirkin (2013) where the effect was attributed to reduced Mie scattering along this flow. The observed sign is qualitatively also consistent with an absorption anisotropy.

The amplitude of the intensity modulation increases with depth. This is in part likely caused by the strengthening of the girdle fabric as well as the strong reduction in overall scattering, as bubbles are transformed to craigite.

To be able to disentangle these two effect, and to potentially transition from the presented experimental ratios to a quantitative measurement of fabric properties, will require a full photon propagation simulation incorporating both Mie scattering on impurities as well as the diffusion introduced through the polycrystalline, birefringent fabric. While the basics for such a simulation have been outlined by Chirkin and Rongen (2019), more work will be required for the simulation to be efficient enough to be used in this application. It is currently also unclear if the intensity ratio alone, will be sufficient to constrain the different fabric properties, namely the Woodcock parameters and the average grain size and shape, or if more information such as timing may be required.

*Competing interests.* The authors declare that they have no competing interests (both financial or non-financial).

*Author contributions.* RB designed the dust logger. The logging was carried out by RB and SB. MR and RB developed the data processing and analysis. The manuscript was prepared by MR with contributions from all co-authors.

*Acknowledgements.* The authors would like to thank the SPICEcore collaboration for providing the borehole, the U.S. Ice Drilling Program, the Antarctic Support Contractor and the NSF National Science Foundation for providing the equipment to perform the described measurement and for their support at South Pole. This work has been achieved under the NSF grant #1443566 and was in part supported by BMBF, Verbundforschung. We would also like to thank the IceCube collaboration for supporting the 18/19 logging activities.





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
