# Peer review of "Observation of an optical anisotropy in the deep glacial ice at the geographic South Pole using a laser dust logger"

_The Cryosphere, 2020_

## Referee Comment (RC1) · Anonymous Referee #1 · 3 Apr 2020

General:

The authors present observation on anisotropic light scattering in the ice sheet near South Pole. The results are derived from measurements with an optical dust logger deployed in a 1750-m borehole. The authors attribute the anisotropy to a preferential light diffusion by the birefringent property of ice, which could be the basis for a new technique for in situ logging of ice fabric properties. The paper is a clear description and analysis of the novel observations.

Specific comments:

p. 2, l. 16: "For crystal realizations where the deflection outweighs the additional

diffusion, the photon flux along the flow axis will continuously increase with distance". I suggest expanding the introduction here: What is additional diffusion? What means photon flux increases with distance? Probably it (finally) decreases by absorption and scattering.

p. 4, l. 7: "anti-freeze": general definition of an anti-freeze is an additive that lowers the freezing point of a water-based liquid. Here, Estisol-140 is a non-freezing drilling fluid.

p. 4, l. 9+13: Voigt (2017) is not in references.

p. 4, l. 11: The accumulation site varies with the depth below the surface. Titan Dome is the accumulation site only for the deepest ice

p. 6, l. 10: "craigite" was suggested as mineral name in the 1980's but to my knowledge never approved. Use clathrate hydrate.

p. 6, l. 15 "global offsets". As the logs are from the same hole, the "global" offsets do not have an external origin. They are rather caused by the logging instrument(?) If so, the term "global" seems inadequate to me.

---

## Referee Comment (RC2) · Jan Eichler (Referee) · 27 Apr 2020

**General comments**

The manuscript describes observations of direction-dependent intensities of back-scattered light in ice. The scattering anisotropy was measured using a laser dust logger deployed in the SPICEcore borehole. The study follows previous observations of anisotropic light propagation in ice at the IceCube observatory. The measurements support results obtained from simulations of light diffusion by Chirkin and Rongen (2019). The authors conclude that scattering due to reflection and refraction on grain

boundaries in a birefringent polycrystal play a major role for the anistropic modulation of light in ice.

The study is carefully carried out and well presented. It shows novel in-situ measurements of optical properties of ice. I believe it is a relevant contribution to our understanding of light propagation and scattering in ice. It may find application in current ice-core-analytical methods as well as in future developments.

I have some concerns about the general suitability of the method for obtaining a continuous fabric record. The intensity ratios can serve to determine the strength of anisotropy within a horizontal plane, so it works well for a girdle-type fabric. However, other types of arrangements of the c-axes can hardly be distinguished or even detected using the current design of the dust logger (e. g. a vertical single maximum, or other more complex distributions).

**Specific comments**

- p.1 l.7-8: See the last paragraph in general comments. I think the statement is too brave. Maybe it is just the term "fabric" which, in my understanding, includes a number of characteristics, which I am not convinced they are accessible for the dust logger.

- p.2 l.16-17: What crystal realization would strengthen the deflection effect?

- p.4 l.11-12: The corresponding accumulation site varies with depth, so the Titan Dome must refer to a particular depth in the ice core.

- p.6 l.10 and p.10 l.8: "craigite": The common terms are clathrate hydrates or air hydrates.
- p.9 Fig.7: If there is no anisotropy signature above 1100 m, how can you determine the anisotropy axis being 120 degrees above this depth?

- p.10 l.10-16: I also wonder what would be the effect of the grain shape and its preferred orientation. Also the scattering on grain boundary intersections (triple junctions) may add some contribution - possibly even an anisotropic one.

---

## Author Comment (AC1) · 15 May 2020

Dear Referee,

Thank you for your timely review. Please find the responses to the issues raised in-line with your review comments below:

General:

[Figure]

The authors present observation on anisotropic light scattering in the ice sheet near South Pole. The results are derived from measurements with an optical dust logger deployed in a 1750-m borehole. The authors attribute the anisotropy to a preferential light diffusion by the birefringent property of ice, which could be the basis for a new technique for in situ logging of ice fabric properties. The paper is a clear description and analysis of the novel observations.

Specific comments:

p. 2, l. 16: "For crystal realizations where the deflection outweighs the additional diffusion, the photon flux along the flow axis will continuously increase with distance". I suggest expanding the introduction here: What is additional diffusion? What means photon flux increases with distance? Probably it (finally) decreases by absorption and scattering.

> True. The absolute photon flux does decrease with distance. But it increases with respect to the intensity on the orthogonal axis.
> The additional diffusion was also meant with this comparison in mind, comparing the larger diffusion along the flow axis with the smaller diffusion along the orthogonal axis.
>
> The paragraph has been expanded to read as follows:
> *The deflection per distance increases for stronger girdle fabrics, a larger average crystal elongation or a smaller average crystal size. For crystal realizations where the deflection outweighs the additional diffusion along the flow axis compared to the diffusion along the orthogonal direction, the photon flux along the flow axis*

*will increase with distance compared to the photon flux along the
orthogonal axis.*

p. 4, l. 7: "anti-freeze": general definition of an anti-freeze is an additive that lowers
the freezing point of a water-based liquid. Here, Estisol-140 is a non-freezing drilling
fluid.

Thank you for the clarification. This has been adopted.

p. 4, l. 9+13: Voigt (2017) is not in references.

That reference had an error and was cited by the first name, not the
family name. This has been corrected.

.p. 4, l. 11: The accumulation site varies with the depth below the surface. Titan Dome
is the accumulation site only for the deepest ice.

That's a good point. The sentence has been changed to:
*The associated accumulation site for the deepest ice is believed by
**?** to be Titan Dome, meaning that the ice has been transported as
far as 200 km.*

p. 6, l. 10: "craigite" was suggested as mineral name in the 1980's but to my

knowledge never approved. Use clathrate hydrate.

Done.

p. 6, l. 15 "global offsets". As the logs are from the same hole, the "global" offsets do not have an external origin. They are rather caused by the logging instrument(?) If so, the term "global" seems inadequate to me.

We have expanded on the explanation. The sentence now reads as:
*These systematic offsets, caused for example by the changing clarity of the drilling fluid or grime accumulation on the logger, are corrected using a second-degree polynomial fitted to the ratio.*

---

## Author Comment (AC2) · 15 May 2020

Dear Jan Eichler,

Thank you for your timely review. Please find the responses to the issues raised in-line with your review comments below:

**General comments:**
The manuscript describes observations of direction-dependent intensities of backscattered light in ice. The scattering anisotropy was measured using a laser dust logger deployed in the SPICEcore borehole. The study follows previous observations of anisotropic light propagation in ice at the IceCube observatory. The measurements support results obtained from simulations of light diffusion by Chirkin and Rongen (2019). The authors conclude that scattering due to reflection and refraction on grain boundaries in a birefringent polycrystal play a major role for the anistropic modulation of light in ice.

The study is carefully carried out and well presented. It shows novel in-situ measurements of optical properties of ice. I believe it is a relevant contribution to our understanding of light propagation and scattering in ice. It may find application in current ice-core-analytical methods as well as in future developments.

I have some concerns about the general suitability of the method for obtaining a continuous fabric record. The intensity ratios can serve to determine the strength of anisotropy within a horizontal plane, so it works well for a girdle-type fabric. However, other types of arrangements of the c-axes can hardly be distinguished or even detected using the current design of the dust logger (e. g. a vertical single maximum, or other more complex distributions).

**Specific comments:**

p.1 l.7-8: See the last paragraph in general comments. I think the statement is too brave. Maybe it is just the term "fabric" which, in my understanding, includes a number of characteristics, which I am not convinced they are accessible for the dust logger.

We agree with the assessment, that the current data will most likely

not be able to unambiguously constrain the many fabric properties. The requirements to study this possibility (photon propagation simulation of the experimental setup), as well as potential improvements to the experimental data (to include the measurement of propagation delays), are detailed in the last paragraph of section 6.

The sentence in the abstract has been changed to:
*In the future, the measurement principle, when combined with a full-chain simulation, may have the potential to provide a continuous record of fabric properties along the entire depth of a drill hole.*

p.2 l.16-17: What crystal realization would strengthen the deflection effect?

A stronger deflection effect would result from a stronger girdle, a larger mean crystal elongation as well as a smaller overall crystal size.

The paragraph has been expanded to read as follows (also taking into account comments from the other reviewer):
*The deflection per distance increases for stronger girdle fabrics, a larger average crystal elongation or a smaller average crystal size. For crystal realizations where the deflection outweighs the additional diffusion along the flow axis compared to the diffusion along the orthogonal direction, the photon flux along the flow axis will increase with distance compared to the photon flux along the orthogonal axis.*

p.4 l.11-12: The corresponding accumulation site varies with depth, so the Titan Dome must refer to a particular depth in the ice core.

That's a good point (as also raised by the other reviewer). The sentence has been changed to:
*The associated accumulation site for the deepest ice is believed by* **?** *to be Titan Dome, meaning that the ice has been transported as far as 200 km.*

p.6 l.10 and p.10 l.8: "craigite": The common terms are clathrate hydrates or air hydrates.

The term has been changed to "clathrate hydrate".

p.9 Fig.7: If there is no anisotropy signature above 1100 m, how can you determine the anisotropy axis being 120 degrees above this depth?

Thank you very much for raising this issue! We had originally assumed that the correctly fitted axis might be an indication for a small residual anisotropy signature. This is not the case. We have instead traced the problem to a minimizer artifact. The seed value for the angle was originally chosen to be $120°$. In the absence of a significant signal, the fit values seem to be pulled towards the seed value instead of scattering randomly. Re-seeding the minimizer with different anisotropy axis directions did not significantly change the measured anisotropy strength or axis below 1100 m (within the uncertainty).

The original paragraph read as:
*Individual ratios are seen to yield consistent anisotropy axes in the deep ice. In the shallow ice above ~1100 m, where the mean strength of the observed anisotropy signal vanishes, the phase angle is badly constrained and varies widely between individual ratios. The per-depth average axes are compatible with the global average of $126 \pm 1$ (stat)$^\circ$.*

This will be changed to:
*Individual ratios are seen to yield consistent anisotropy axes in the deep ice. In the shallow ice above ~1100 m, where the mean strength of the observed anisotropy signal vanishes, the phase angle is unconstrained. The average axis in the deep ice is $126 \pm 1$ (stat)$^\circ$.*

Attached please find a figure showing the impact of the seed value as well as the plot to be used in the updated version of the manuscript.

p.10 l.10-16: I also wonder what would be the effect of the grain shape and its preferred orientation. Also the scattering on grain boundary intersections (triple junctions) may add some contribution - possibly even an anisotropic one.

Grain shape after averaging over many thousand encountered crystals is by us assumed to be an ellipsoid with the mayor axis being closely aligned with the flow direction. It is the second major contribution (besides the c-axis distribution) to the effect, as it

changes the grain boundary orientation distribution. As shape and fabric are strongly linked (in a girdle fabric the average crystal shape has to be elongated) we did not explicitly elaborate on this. We hope this is now covered, by the extended description of the general birefringence effect on page 2.

Triple junctions have so far not been considered. But as their fractional contribution to the surface area of a grain is rather small, their effect should be negligible.

[Figure]

**Fig. 1.** Impact of the minimizer seed on the axis fit.

[Figure]

**Fig. 2.** New figure for the depth dependent anisotropy axis.

---

## Author Response (ED1)

Dear Referee,

Thank you for your timely review. Please find the responses to the issues raised in-line with your review comments below:

General:

The authors present observation on anisotropic light scattering in the ice sheet near South Pole. The results are derived from measurements with an optical dust logger deployed in a 1750-m borehole. The authors attribute the anisotropy to a preferential light diffusion by the birefringent property of ice, which could be the basis for a new technique for in situ logging of ice fabric properties. The paper is a clear description and analysis of the novel observations.

Specific comments:

p. 2, l. 16: "For crystal realizations where the deflection outweighs the additional diffusion, the photon flux along the flow axis will continuously increase with distance". I suggest expanding the introduction here: What is additional diffusion? What means photon flux increases with distance? Probably it (finally) decreases by absorption and scattering.

True. The absolute photon flux does decrease with distance. But it increases with respect to the intensity on the orthogonal axis.
The additional diffusion was also meant with this comparison in mind, comparing the larger diffusion along the flow axis with the smaller diffusion along the orthogonal axis.

The paragraph has been expanded to read as follows:
*The deflection per distance increases for stronger girdle fabrics, a larger average crystal elongation or a smaller average crystal size. For crystal realizations where the deflection outweighs the additional diffusion along the flow axis compared to the diffusion along the orthogonal direction, the photon flux along the flow axis*

*will increase with distance compared to the photon flux along the
orthogonal axis.*

p. 4, l. 7: "anti-freeze": general definition of an anti-freeze is an additive that lowers
the freezing point of a water-based liquid. Here, Estisol-140 is a non-freezing drilling
fluid.

Thank you for the clarification. This has been adopted.

p. 4, l. 9+13: Voigt (2017) is not in references.

That reference had an error and was cited by the first name, not the
family name. This has been corrected.

.p. 4, l. 11: The accumulation site varies with the depth below the surface. Titan Dome
is the accumulation site only for the deepest ice.

That's a good point. The sentence has been changed to:
*The associated accumulation site for the deepest ice is believed by
? to be Titan Dome, meaning that the ice has been transported as
far as 200 km.*

p. 6, l. 10: "craigite" was suggested as mineral name in the 1980's but to my

knowledge never approved. Use clathrate hydrate.

Done.

p. 6, l. 15 "global offsets". As the logs are from the same hole, the "global" offsets do
not have an external origin. They are rather caused by the logging instrument(?) If so,
the term "global" seems inadequate to me.

We have expanded on the explanation. The sentence now reads as:
*These systematic offsets, caused for example by the changing clarity
of the drilling fluid or grime accumulation on the logger, are corrected
using a second-degree polynomial fitted to the ratio.*

[Figure]

Thank you for your timely review. Please find the responses to the issues raised in-line with your review comments below:

**General comments:**
The manuscript describes observations of direction-dependent intensities of backscattered light in ice. The scattering anisotropy was measured using a laser dust logger deployed in the SPICEcore borehole. The study follows previous observations of anisotropic light propagation in ice at the IceCube observatory. The measurements support results obtained from simulations of light diffusion by Chirkin and Rongen (2019). The authors conclude that scattering due to reflection and refraction on grain boundaries in a birefringent polycrystal play a major role for the anistropic modulation of light in ice.

The study is carefully carried out and well presented. It shows novel in-situ measurements of optical properties of ice. I believe it is a relevant contribution to our understanding of light propagation and scattering in ice. It may find application in current ice-core-analytical methods as well as in future developments.

I have some concerns about the general suitability of the method for obtaining a continuous fabric record. The intensity ratios can serve to determine the strength of anisotropy within a horizontal plane, so it works well for a girdle-type fabric. However, other types of arrangements of the c-axes can hardly be distinguished or even detected using the current design of the dust logger (e. g. a vertical single maximum, or other more complex distributions).

**Specific comments:**

p.1 l.7-8: See the last paragraph in general comments. I think the statement is too brave. Maybe it is just the term "fabric" which, in my understanding, includes a number of characteristics, which I am not convinced they are accessible for the dust logger.

We agree with the assessment, that the current data will most likely

not be able to unambiguously constrain the many fabric properties. The requirements to study this possibility (photon propagation simulation of the experimental setup), as well as potential improvements to the experimental data (to include the measurement of propagation delays), are detailed in the last paragraph of section 6.

The sentence in the abstract has been changed to:
*In the future, the measurement principle, when combined with a full-chain simulation, may have the potential to provide a continuous record of fabric properties along the entire depth of a drill hole.*

p.2 l.16-17: What crystal realization would strengthen the deflection effect?

A stronger deflection effect would result from a stronger girdle, a larger mean crystal elongation as well as a smaller overall crystal size.

The paragraph has been expanded to read as follows (also taking into account comments from the other reviewer):
*The deflection per distance increases for stronger girdle fabrics, a larger average crystal elongation or a smaller average crystal size. For crystal realizations where the deflection outweighs the additional diffusion along the flow axis compared to the diffusion along the orthogonal direction, the photon flux along the flow axis will increase with distance compared to the photon flux along the orthogonal axis.*

p.4 l.11-12: The corresponding accumulation site varies with depth, so the Titan Dome must refer to a particular depth in the ice core.

That's a good point (as also raised by the other reviewer). The sentence has been changed to:
*The associated accumulation site for the deepest ice is believed by* **?** *to be Titan Dome, meaning that the ice has been transported as far as 200 km.*

p.6 l.10 and p.10 l.8: "craigite": The common terms are clathrate hydrates or air hydrates.

The term has been changed to "clathrate hydrate".

p.9 Fig.7: If there is no anisotropy signature above 1100 m, how can you determine the anisotropy axis being 120 degrees above this depth?

Thank you very much for raising this issue! We had originally assumed that the correctly fitted axis might be an indication for a small residual anisotropy signature. This is not the case. We have instead traced the problem to a minimizer artifact. The seed value for the angle was originally chosen to be 120°. In the absence of a significant signal, the fit values seem to be pulled towards the seed value instead of scattering randomly. Re-seeding the minimizer with different anisotropy axis directions did not significantly change the measured anisotropy strength or axis below 1100 m (within the uncertainty).

The original paragraph read as:
*Individual ratios are seen to yield consistent anisotropy axes in the deep ice. In the shallow ice above $\sim 1100\,m$, where the mean strength of the observed anisotropy signal vanishes, the phase angle is badly constrained and varies widely between individual ratios. The per-depth average axes are compatible with the global average of $126 \pm 1$ (stat)$^{\circ}$.*

This will be changed to:
*Individual ratios are seen to yield consistent anisotropy axes in the deep ice. In the shallow ice above $\sim 1100\,m$, where the mean strength of the observed anisotropy signal vanishes, the phase angle is unconstrained. The average axis in the deep ice is $126 \pm 1$ (stat)$^{\circ}$.*

Attached please find a figure showing the impact of the seed value as well as the plot to be used in the updated version of the manuscript.

p.10 l.10-16: I also wonder what would be the effect of the grain shape and its preferred orientation. Also the scattering on grain boundary intersections (triple junctions) may add some contribution - possibly even an anisotropic one.

Grain shape after averaging over many thousand encountered crystals is by us assumed to be an ellipsoid with the mayor axis being closely aligned with the flow direction. It is the second major contribution (besides the c-axis distribution) to the effect, as it

changes the grain boundary orientation distribution. As shape and fabric are strongly linked (in a girdle fabric the average crystal shape has to be elongated) we did not explicitly elaborate on this. We hope this is now covered, by the extended description of the general birefringence effect on page 2.

Triple junctions have so far not been considered. But as their fractional contribution to the surface area of a grain is rather small, their effect should be negligible.

[Figure]

**Fig. 1.** Impact of the minimizer seed on the axis fit.

[Figure]

**Fig. 2.** New figure for the depth dependent anisotropy axis.

[revised manuscript text omitted]

---

## Author Response (AR2)

Dear Olaf Eisen,

thank you for taking another detailed look at the manuscript.
Below please find detailed responses to some of the questions you raised, as well as a new marked-up version of the manuscript.

> Dear Martin & coauthors,
>
> thank you for your revision, in which you addressed most of the criticism raised by the reviewers. In the attached pdf I made a number of small corrections which I ask you to incorporate.

Most all comments were incorporated, although we did not quite know how you want to see the sentence on page 6 line 55ff ("To be able to disentangle these two effects,...") to be changed.

The word "timing" was indeed not very descriptive and has been substituted with "distribution of the propagation delay of individual photons". Individual photons can be time-stamped with nanosecond precision using for example photomultipliers.
For details of that method please see for example doi:10.1029/2005JD006687.

On page 6, line 45 we are indeed talking about an absorption based anisotropy in contrast to the more general attenuation based anisotropy. Attenuation includes effects from both absorption and scattering. As mentioned in the manuscript, modelling the anisotropy through either absorption or scattering is badly motivated (for details see 10.18154/RWTH-2019-09941), but both approaches can result in an effective description of the effect seen in IceCube. In both cases less scattering or absorption is required for photons travelling along the flow axis to match the IceCube measurement. In the case of the dustlogger this would result in a larger return intensity in the case of the absorption (as also observed) and a smaller return signal in the case of the scattering model.

> One point I would ask you to clarify is the comment and your respond on page C5 by Jan Eichler: "p.10 l.10-16: I also wonder what ..."
>
> In principle this is correct, with a few constrains. I would appreciate if you would elaborate on this a little bit further on this in the manuscript (although you don't have to go into the details of fabric/grain development, which are not the main focus in your manuscript):
>
> - For a girdle, the connection of mean grain elongation to be perpendicular to fabric has been described early, I think Gow. Could you add a reference? This is only true, to my understanding, for the right ratio of deformation and recrystallization, e.g. fast and/or cold.

- "assumed to be an ellipsoid" To you have any reference/data to this?

We know from private communication of yet unpublished material from SP14 that the average elongation is indeed aligned with the girdle normal vector, as one would expect from a simple deformation scenario. (This we can obviously not add to the manuscript.)

Averaging over an ensemble of randomly shaped polyhedra with an average elongation will always result in an ellipsoid. That the ellipsoid well describes the properties of the ensemble of polyhedra has for example for the surface orientation density been studied in 10.18154/RWTH-2019-09941 by comparing the ellipsoid expectation to surface orientations from crystal tessellation simulations.

In general we believe that adding the explanations given above would clutter the manuscript. From the current explanations it should be clear that both grain shape, orientation and fabric can have an effect on the propagation of light. As neither the current data, nor the current simulation offer the possibility to distinguish the different crystal properties, we prefer to leave these details for for a future paper.

Please provide a reference for your azimuth (i.e. where is 0° pointing to?) - I did not see that defined.

Done. The azimuth is with respect to Local Grid bearings. Grid North (0°) aligns with the Greenwich meridian.

At some point you mention "water filled borehole". I guess that is a typo for SP?

We are not quite sure what SP stands for in this context. But that is surprisingly not a typo. As part of the construction of the IceCube Neutrino Observatory (2004-2010), 86 2500 m deep, 60 cm diameter holes were drilled using hot water drilling (see doi:10.3189/2014AoG68A03). These holes took about 2 days to drill each and required two to three weeks to fully refreeze. This time not only allowed us to instrument the holes with our photosensors, but also left some time for logger deployments.

Are the data available for others, according to TC guidelines, e.g. in a data repository?

There is currently one data release available at https://doi.org/10.15784/601222, for which we have added a data availability section in the manuscript. It includes a single pre-processed stratigraphy. In the context of this paper it gives a good idea about the data quality, but can not be used to reproduce the anisotropy signature, as only a single stratigraphy and no heading information is provided. A data release of the full set of logs is currently being discussed, but has not yet been prepared.

[revised manuscript text omitted]

---

## Author Response (AR3)

Dear Olaf Eisen,

thank you for the quick iteration on the minor revision.

> Dear Martin, Ryan and Summer,
> thank your for the replies, which clarified some issues. I'm aware that the holes at SP (south pole) were drilled with hot water, but I would ask you to add a few words, e.g. in the water-filled boreholes (from hot water drilling before refreezing after 2-3 days).

Thank you for the suggestion. We added
"(from hot water drilling and before completely refreezing after 2-3 weeks)" to the text.

> Please add the private communication as such (e.g. NAME, personal communication, 2019). Then it is obvious where the info is coming from.

As detailed in the previous response we feel that adding the section, to which this citation would be relevant, *"would clutter the manuscript. From the current explanations it should be clear that both grain shape, orientation and fabric can have an effect on the propagation of light. As neither the current data, nor the current simulation offer the possibility to distinguish the different crystal properties, we prefer to leave these details for for a future paper."* As such the private communication was not incorporated.

> Data repository: it is not a requirement, but it is greatly appreciated in the context of open science and the guidelines to make your data set publicly available. Moreover, data sets (citations, downloads) are becoming another metrics for science evaluation, so it is an opportunity for you, too.

With the previous iteration we added the following data availability section:
*"A single stratigraphy is currently available from https://doi.org/10.15784/601222 (Bay, R. (2019)). The full set of logs may be released in the future."*

Best regards
Summer, Ryan & Martin